

# Transcriptomics analysis reveals the effect of *Broussonetia papyrifera L.* fermented feed on meat quality traits in fattening lamb

Xuejiao An[1], Shengwei Zhang[2], Taotao Li[1], Nana Chen[1], Xia Wang[1], Baojun Zhang[2] and Youji Ma[1]

[1] College of Animal Science and Technology, Gansu Agricultural University, Lanzhou, Gansu, China
[2] Gansu Provincial Farmer Education and Training Station, Lanzhou, China

## ABSTRACT

To date, utilization of feed grains is increasing, which competes for human food. It is imperative to develop and utilize unconventional feed materials. *Broussonetia papyrifera L.* (*B. papyrifera*) is a good feeding material with high crude protein, crude fat, and low crude fiber, which is widely distributed in China. In this study, 12 Dorper ♂ × Hu ♀ crossbred weaned male lambs were seleted into four groups based on the feed that ratio of the *B. papyrifera* fermented feed in the total mixed diet (0%, 6%, 18%, and 100%), to character the lambs' longissimus dorsi (LD) fatty acids, morphology and transcriptome. Results showed that the muscle fiber's diameter and area were the smallest in the 100% group. The highest content of beneficial fatty acids and the lowest content of harmful fatty acids in group 18%. RNA-seq identified 443 differentially expressed genes (DEGs) in the LD of lambs from 4 groups. Among these genes, 169 (38.1%) were up-regulated and 274 (61.9%) were down-regulated. The DEGs were mostly enriched in in fatty acid metabolism, arginine and proline metabolism, and PPAR signaling pathways. Our results provide knowledge to understand effect of different ratios of *B. papyrifera* fermented feed on sheep meat quality traits, also a basis for understanding of the molecular regulation mechanism of *B. papyrifera* fermented feed affecting on sheep meat quality.

## INTRODUCTION

*Broussonetia papyrifera L.* (*B. papyrifera*), is a broad-leaved woody specie with a deciduous, dioecious, and dichogamous plant (*Hong, Yang & Liao, 2009*). It is native to southern China and Japan, and now distributes in China, Malay Peninsula, Japan, and Pacific islands (*Zerega et al., 2005; Liu, Fan & Shen, 2009*). The plant *B. papyrifera* has strong stress resistance, can grow normally at arid hillside, valley, and roadside. Studies have been indicated thet it has multiple function, such as the role of manufacturing paper with its inner bark (*Suleman, 1995*), its leaves are ideal for feeding animals (*Si et al., 2018*) and it can also be used in the pharmaceutical industry (*Ko et al., 2011*). Domestic and foreign scholars have found that *B. papyrifera* contains a large amount of flavonoids and diphenylpropane compounds,

Corresponding author
Youji Ma, yjma@gsau.edu.cn

which have certain antioxidant properties. And the mulberry tree contains at least 16 kinds of amino acids, of which seven are essential amino acids. And its protein content is rich, the total amino acid content is up to 24.35%, it is a good feed material (*Wei, Liu & Wan, 2008*). It is found that *B. papyrifera* silage also reduce the ruminal biohydrogenation (*Yusuf et al., 2017*) and increase the poly-unsaturated fatty acids (PUFA) concentration in the milk (*Si et al., 2018*).

There are about 300,000 hectares of *B. papyrifera* in China, widely distributes in basins of Yellow river, Yangtze river, Pearl river and Mingjiang river. *B. papyrifera* is a kind of nutrient-rich woody feed. It contains high crude protein, crude ash, crude fat, phosphorus, and suitable crude fiber content. It has the potential to alleviate the lack of protein feed in China and its dependence on foreign sources. It was reported that the crude protein of *B. papyrifera* had high degradation rate in the rumen of cows. Therefore, this kind of unconventional woody forage was considered to be of great development value (*Wang et al., 2019*). Also the *B. papyrifera* fruits polysaccharides have antioxidant and antibacterial activities (*Roth & Wolfenson, 2016*). Many of the plants contain phytochemicals, which had potent antioxidant activities (*Lee et al., 1991*; *Conforti et al., 2008*; *Xu et al., 2010*; *Roth & Wolfenson, 2016*). Antioxidant activities have been described for related polyphenolic constituents extracted from the stem, bark and wood of *B. papyrifera* (*Xu et al., 2010*). There is report about the radix of *B. papyrifera* that had the greatest antinociceptive and anti-inflammatory effects when different parts of the plant were compared as treatment for chemical-induced pain and inflammation in rodents (*Hong et al., 2013*).

With the improvement of people's living standards, consumers' demand for livestock and poultry products has undergone great changes. People not only demand good taste and flavor, but also demand rich nutrition and benefit human health (*Lin et al., 2014*). Due to its high protein content, rich nutrient content, and low fat and cholesterol content, mutton has a growing demand for mutton. The main indicators for comprehensively evaluating individual animal meat production performance and meat nutritional value include meat production, meat quality, muscle nutrition and fatty acid content. The important indicators for evaluating meat quality are tenderness, marble pattern, meat color, etc. (*Sun, 2017*). At the same time, the composition and content of fatty acids in muscle are of great significance to meat flavor and human health (*Zhang et al., 2019*, *Guo et al., 2019*). Changing in meat quality traits are regulated by related genes. Therefore, functional genes that affect sheep meat quality can be screened through transcriptome sequencing. At the same time, adding different ratios of *B. papyrifera* fermented feed to sheep diets can affect meat quality related genes the expression changes to affect sheep meat quality traits. *Cao (2017)* used Dorper ×Small Tail Han Crossbred and Small Tail Han Sheep as research objects to compare and analyze their longissimus dorsi transcriptome sequencing and screened out 16 functional genes that may affect sheep meat quality traits. *Hao, Cui & Gu (2016)* used RNA-seq and DNA methylation differential gene joint analysis to obtain a large number of genes related to muscle development, muscle meat traits, muscle energy and lipid metabolism, and cellular defense and stress response. In light of this, the different ratio of *B. papyrifera* fermented feed was used as protein feed to feed fattening lambs. The fatty acid content and haematoxylin and eosin (HE) staining was measured to observe the longissimus dorsi

(LD) phenotype, to study the feeding effect of *B. papyrifera* fermented feed; then, the transcriptome data was alanyzed, to screen differential genes that relate to *B. papyrifera* fermented feed. Conjoint analysis explored the influence of *B. papyrifera* fermented feed on the quality of lamb meat, and screened out the appropriate addition ratio.

## MATERIAL AND METHODS

### Overview of experimental program

This paper mainly explores the molecular regulation mechanism of different ratios of *B. papyrifera* fermented feed on sheep meat quality. We randomly selected 12 crossbred weaned male lambs divided into four groups (three lambs per group) fed that ratio of the *B. papyrifera* fermented feed in the total mixed diet (0%, 6%, 18%, and 100%) for 60 days. HE staining and analysis of fatty acid content of LD muscle, a suitable ratio of fermented feed for *B. papyrifera* was initially selected. Then conduct transcriptomics analysis to screen the key genes that affect meat quality and predict the biological functions of these genes in fatty acid synthesis and metabolism. Finally, combining the phenotypic data and the transcriptome results, explore the effect of *B. papyrifera* fermented feed on the quality of lamb meat, and select the appropriate addition ratio (Fig. 1).

### Chemicals and reagents

NaOH (Jinhuitaiya, Tianjin), $CH_3OH$ (jingke, Wuxi), TRIpure Reagent (DiNing Biotech, China), $CHCl_3$ (Beyotime), NaCl (Jingke, Wu Xi), 14%BF3-CH3OH (Xiya, Shandong), Eosin Staining Solution (Beyotime), Hematoxylin Staining Solution (Beyotime), *Evo M-MLV* RT Kit with gDNA Clean for qPCR (Accurate Biotechnology (Hunan) Co., Ltd), YBR Green Premix *Pro Taq* HS qPCR Kit (Accurate Biotechnology (Hunan) Co., Ltd).

### Ethics approval

Experimental animals were reviewed and approved by the Animal Committee of Gansu Agricultural University (GSAU-2019-76).

### Preparation of experimental animals

12 crossbred weaned male lambs Dorper ♂ × Hu ♀ were selected, which with same weight (20 kg/lamb), similar ages and good health. All fed in Gansu Zhongtian Sheep Industry Co., Ltd. (Longxi, China). Randomly divided into four groups (three lambs per group) based on the feed that ratio of the *B. papyrifera* fermented feed in the total mixed diet (0%, 6%, 18%, and 100%). There is transition period of 7 days, a pre-test period of 7 days and a formal period of 60 days. Refer to NRC (2007) standard for 20 kg lamb with a daily weight gain of 200g/d. The diet formula was shown in Table 1. Feed the same amount twice a day at 08:00 and 18:00, respectively, and drink freely. 12 lambs were euthanized after 60 days of feeding each lamb was given intravenous injection of 360 mg sodium pentobarbital without heartbeat, continuous non-spontaneous breathing for 2–3 min, and no blinking reflex, and then dissected. Collecting the LDs: one sample was immediately placed in liquid nitrogen and subsequently cryopreserved at −080 °C for the extraction of total RNA, the second was placed in an ice box and subsequently cryopreserved at −20 °C to extract fatty acids, and the last was fixed with 4% paraformaldehyde for approximately 48 h, dehydrated

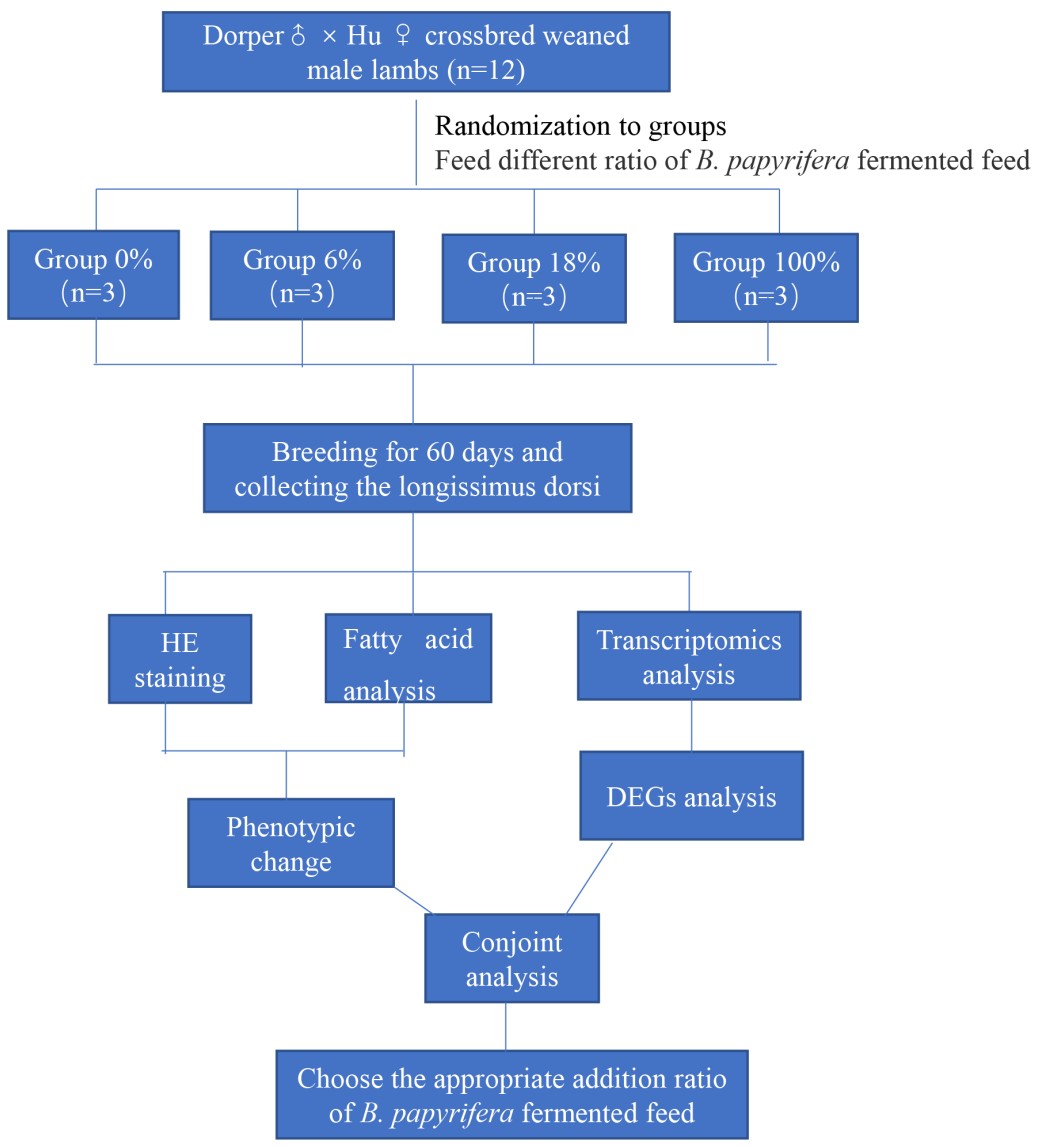

**Figure 1  Experimental design and procedures.**

in a gradient series of ethanol, cleared inxylene and embedded in paraffin. The tissues were sectioned at five mm thickness and used for HE staining.

## Fatty acid analysis

Frozen samples were thawed 12 h prior to analyse at 4 °C, and weighed 5g LD muscle on an electronic balance. In order to accurately and quickly extract the lipids in the LD, total lipids were extracted with $CHCl_3/CH_3OH$ (2:1, V/V) as described by *Zlatkis, Zak & Boyle (1953)* with some modifications. Fatty acid methyl esters (FAME) were prepared as following from the methods as described previously (*Liu , 2015*) with some modification. Extracted lipids (approximately 10 mg) were combined with BHT and 2.0 mL NaOH-MeOH solution (0.1 mol/L), saponify at 60 °C for 60 min. After the solution was clear and transparent, test

**Table 1 Composition and nutrient levels of experimental diets (DM basis) %.** The Premix Provided the following per kg of diets: VA 220000 IU, VD3 72000 IU, VE 2000 IU, D-biotin 40.0 Mixed group, nicotinic acid amide 2000 Mixed group, Mn (as manganese sulfate) 710 Mixed group, Zn (as zinc sulfate) 2005 Mixed group, Fe (as ferrous sulfate) 830.0 Mixed group, Cu (as copper sulfate) 680.0 Mixed group, Go (as Cobalt sulfate) 12 Mixed group.

| Items | 0% | 6% | 18% | 100% |
|---|---|---|---|---|
| Ingredients | | | | |
| B. papyrifera fermented feed | 0.0 | 6.0 | 18.0 | 100 |
| Corn silage | 45.7 | 39.3 | 22.5 | 0.0 |
| Corn | 16.6 | 21.5 | 36.8 | 0.0 |
| Soybean meal | 26.7 | 23.5 | 17.7 | 0.0 |
| Bran | 8.1 | 7.3 | 3.3 | 0.0 |
| Limestone | 1.2 | 0.7 | 0.00 | 0.0 |
| NaCl | 0.7 | 0.7 | 0.7 | 0.0 |
| Premix 1% | 1.0 | 1.0 | 1.0 | 1.0 |
| Total | 100.0 | 100.0 | 100.0 | 100.0 |
| Nutrient levels | | | | |
| DE (MJ/kg) | 10.38 | 10.35 | 10.36 | 8.13 |
| CP | 17.42 | 17.40 | 17.40 | 26.10 |
| Ca | 20.50 | 19.47 | 15.95 | 15.90 |
| TP | 11.06 | 10.58 | 8.9 | 13.00 |
| NDF | 0.60 | 0.61 | 0.72 | 3.40 |
| ADF | 0.34 | 0.33 | 0.30 | 0.20 |

the tube under 4 °C running water cool to room temperature; then add 2.0 mL of 14% $BF_3$-$CH_3OH$ solution, methylate at 60 °C for 15 min, and cool the test tube to RT under 4 °C of tap water; add three mL of n-hexane, one mL of 25% NaCl solution and six mL of distilled water was shaken vigorously, centrifuged at 6,000 r/min for 10 min, the upper layer of FAME was collected, the lower layer solution was repeatedly extracted once with n-hexane, combined FAME, added an appropriate amount of anhydrous sodium sulfate, and all were sucked out with a disposable needle, and the organic phase was filtered. The membrane is placed at the front of the needle tube for filtration, concentrated to dryness under the protection of nitrogen, the residue is dissolved in n-hexane and fixed to a 2.0 mL volumetric flask, stored at −20 °C for future use, until GC detection.

Derivatized methyl esters of fatty acids were separated and quantified by gas chromatograph (7820A; Aglilent Technologies, California, USA). The capillary column was a 100 m ×0.25 mm ×0.2 μm (SPTM-2560; Sigma Louis, MO, USA). The carrier gas (Nitrogen) flow rate was 1.5 mL min$^{-1}$, The flow rate of gas (Hydrogen) was 40 mL min$^{-1}$, The flow rate of air was 400mL min$^{-1}$. Split injection, split ratio 60:1, injection volume 1.0 μL. Injector temperature was set as 260 °C. The initial oven temperature was programmed at 140 °C and maintained for 5 min, then increased to 200 °C at 2 °C min$^{-1}$, increased to 230 °C at 6 °C/min and keep 20 min.

### Haematoxylin and eosin staining

Section of LD of was stained with H&E as described previously (*Asuka et al., 2017*). Then was observed under a microscope, and photographed using ImageView.

### Muscle fiber measurement

The HE-stained section was photographed under the $20\times$ and $40\times$ microscope using the ImageView microphase system, and the diameter and area of the muscle fibers under the $40\times$ microscope were determined using ImageJ software.

### Total RNA extraction, cDNA library preparation

Total RNA was extracted from each sample using TRIpure Reagent according to instructions. The concentrations and quality of RNA samples were examined by NanoDrop2000 (Thermo Fisher, Waltham, MA, USA) and Agilent 2100 (Agilent). The total RNA of the aforementioned samples was used to construct an RNA-Seq library, which were sequenced in parallel on Illumina HiSeq 2000 system by Biomarker Technologies (Beijing, China).

### Identification and quantification of DEGs

Identification of differential genes used FPKM (fragments per kilobase per million reads) (*Mortazavi et al., 2008*). Differential expression analyses of genes among four groups were implemented using the Cuffdiff (*Trapnell et al., 2013*). Genes with indexes fold change (FC) > 1.5 and FDR < 0.05 were considered as DEGs.

### Functional annotation and pathway analysis of DEGs

The functional of DEGs were identified through GO and KEGG database. GO annotation and KEGG pathways analyses by the GOseq R package and KOBAS software, the *p* value $\leq$ 0.05 were defined as significant enriched by DEGs.

### Validation of RNA-seq results

To verify reliability of the transcriptomic profiling data, quantitative real-time PCR (qRT-PCR) was performed for nine randomly selected DEGs (*FABP3*, *ECI1*, *ACADVL*, *GOT1*, *LPIN1*, *MLYCD*, *PLIN5*, *RASD1* and *SELENOW*). *GAPDH* was used as a reference gene in quantitative analysis. Using *Evo M-MLV* RT Kit with gDNA Clean for qPCR to reverse transcribe RNA into cDNA, and using SYBR Green Premix *Pro Taq* HS qPCR Kit and run on the Roche LightCycler96. The relative expression was calculated using the $2^{-\Delta\Delta Ct}$ method (*VanSlyke & Musil, 2000*). The qRT-PCR primers are listed in the electronic supplementary material, Table S1 .

### Statistical analysis

All statistical analyses were conducted with SPSS 22.0 software (SPSS Inc., Chicago, IL, USA), the least significant difference (LSD with Fischer's) method in one-way analysis of variance (ANOVA). The results were expressed as mean $\pm$standard error. *P value* < 0.05 were considered as statistically significant.

**Table 2** Analysis of saturated fatty acid content and composition in LD muscle (g/100g).

| Fatty acid | 0% | 6% | 18% | 100% | *P*-value |
|---|---|---|---|---|---|
| C4: 0 | 1.96 ±0.26 | 2.79 ±0.43 | 2.49 ±0.33 | 3.33 ±0.81 | 0.279 |
| C10: 0 | 0.19 ±0.01 | 0.16 ±0.00 | 0.14 ±0.00 | 0.16 ±0.00 | 0.912 |
| C12: 0 | 0.12 ±0.00b | 0.13 ±0.00b | 0.10 ±0.00b | 0.25 ±0.04a | 0.037 |
| C13: 0 | 0.51 ±0.056 | 0.68 ±0.07 | 0.55 ±0.05 | 0.67 ±0.12 | 0.258 |
| C14: 0 | 2.29 ±0.12b | 2.27 ±0.07b | 2.14 ±0.07b | 2.96 ±0.31a | 0.009 |
| C15: 0 | 0.39 ±0.06bc | 0.47 ±0.05b | 0.30 ±0.03c | 0.66 ±0.06a | 0.000 |
| C16: 0 | 24.21 ±0.34 | 23.82 ±0.35 | 24.18 ±0.28 | 23.63 ±0.81 | 0.693 |
| C17: 0 | 1.14 ±0.09b | 1.00 ±0.04bc | 0.89 ±0.03c | 1.36 ±0.03a | 0.000 |
| C18: 0 | 15.29 ±0.43 | 15.42 ±0.73 | 15.92 ±0.56 | 16.14 ±0.64 | 0.910 |
| C20: 0 | 0.10 ±0.00b | 0.11 ±0.00b | 0.13 ±0.00b | 0.30 ±0.07a | 0.000 |
| C21: 0 | 0.46 ±0.06a | 0.34 ±0.01ab | 0.31 ±0.01b | 0.40 ±0.01b | 0.030 |
| C22: 0 | 0.33 ±0.03a | 0.38 ±0.04a | 0.23 ±0.03b | 0.18 ±0.02b | 0.000 |
| C23: 0 | 2.40 ±0.30 | 2.81 ±0.47 | 2.47 ±0.27 | 2.90 ±0.58 | 0.743 |

**Notes.**
Different lowercase letters in the same industry indicate significant differences ($P < 0.05$).

# RESULTS

## Comparison of differences in fatty acid composition and content among different groups

### Analysis of saturated fatty acid content

A total of 11 kinds of saturated fatty acids were detected in LD of lamb with different *B. papyrifera* fermented feed, including 4 odd saturated fatty acids and 7 even saturated fatty acids. Among the even saturated fatty acids, butyric acid (C4:0), capric acid (C10:0) palmitic acid (C6:0) and stearic acid (C18:0)were not significantly different among the four groups ($P > 0.05$). The content of lauric acid(C12:0), myristic acid (C14:0) and arachidic acid (C20:0) in the 100% group was significantly higher than the other three groups($P < 0.05$). Among the odd saturated fatty acids, tridecanoic acid (C13:0) and tricosanoic acid (C23:0) were not significantly different among the four groups ($P > 0.05$). The content of pentadecanoic acid (C15:0) and heptadecanoic acid (C17:0) group 100% was significantly higher than other groups (Table 2 and Table S2).

### Analysis of unsaturated fatty acid content

A total of 15 kinds of unsaturated fatty acids were detected in LD of lamb fed with different *B. papyrifera* fermented feed, including 7 kinds of MUFA and 8 kinds of PUFA. Among the monounsaturated fatty acids, there are three insignificant differences between the four groups, namely myristic acid (C14:1), palmitoleic acid (C16:1), nervonic acid (C24:1) and trans linoleic acid (C18:2n6t) ($P > 0.05$). Among the unsaturated fatty acids, the content of Cis-10-pentadecenoic acid (C15:1) and cis-10-heptadecenoic acid (C17:1) and were gradually increased with the addition of *B. papyrifera* fermented feed. Among the polyunsaturated fatty acids, the content of four fatty acids also increased with the addition of mulberry fermented feed, including linoleic acid (C18:2n6c), $\gamma$-linolenic acid

**Table 3    Analysis of unsaturated fatty acid content and composition in LD muscle (g/100 g).**

| Fatty acid | 0% | 6% | 18% | 100% | P-value |
|---|---|---|---|---|---|
| C14:1 | 0.24 ±0.05 | 0.31 ±0.05 | 0.30 ±0.04 | 0.34 ±0.07 | 0.600 |
| C15:1 | 0.27 ±0.05c | 0.32 ±0.01b | 0.43 ±0.02a | 0.42 ±0.04a | 0.003 |
| C16:1 | 1.45 ±0.09 | 1.52 ±0.09 | 1.45 ±0.07 | 1.47 ±0.08 | 0.914 |
| C17:1 | 0.50 ±0.02b | 0.55 ±0.01b | 0.64 ±0.03b | 0.97 ±0.31a | 0.042 |
| C18:1n9t | 2.97 ±0.29a | 2.51 ±0.19ab | 2.32 ±0.04b | 2.09 ±0.12b | 0.012 |
| C18:1n9c | 38.30 ±0.69a | 35.49 ±0.55b | 37.85 ±0.27a | 31.5 ±1.16c | 0.000 |
| C20:1 | 0.00 ±0.00b | 0.00 ±0.00b | 0.09 ±0.00a | 0.94 ±0.00a | 0.014 |
| C24:1 | 0.27 ±0.03 | 0.30 ±0.03 | 0.22 ±0.01 | 0.23 ±0.03 | 0.186 |
| C18:2n6t | 0.21 ±0.01 | 0.18 ±0.01 | 0.19 ±0.01 | 0.24 ±0.02 | 0.114 |
| C18:2n6c | 5.67 ±0.35c | 6.25 ±0.62b | 6.28 ±0.60b | 7.15 ±0.50a | 0.000 |
| C18:3n6 | 0.07 ±0.01b | 0.27 ±0.00a | 0.28 ±0.00a | 0.32 ±0.01a | 0.034 |
| C18:3n3 | 0.30 ±0.01b | 0.39 ±0.04b | 0.45 ±0.00b | 2.31 ±0.31a | 0.005 |
| C20:2 | 0.2 ±0.03a | 0.00 ±0.00b | 0.29 ±0.02a | 0.32 ±0.00a | 0.005 |
| C20:3n6 | 0.19 ±0.02a | 0.22 ±0.02a | 0.20 ±0.02a | 0.12 ±0.01b | 0.000 |
| C22:6n3 | 0.18 ±0.01b | 0.21 ±0.02b | 0.27 ±0.02b | 0.64 ±0.12a | 0.000 |

**Notes.**

Different lowercase letters in the same industry indicate significant differences ($P < 0.05$).

(C18:3n6), alpha-linolenic acid (C18:3n3), cis-8, 11,14-eicosatrienoic acid (C20:3n6) and DHA (C22:6n3) (Table 3 and Table S2).

*Analysis of total fatty acid content*

The functional fatty acids that are closely related to human health mainly n3 and n6 PUFAs. This study found that n6 was not significantly different between the four groups ($P > 0.05$), but its content increased with the amount of addition, the content of n3 in the group 100% was significantly higher than other groups ($P < 0.05$). Both n6/n3 and P/S can be used to measure the nutritional value of meat. In this study, P/S was the highest in the 100% group, while n6/n3 was the highest in the group 18% (Table 4 and Table S2).

## Comparison of morphological differences of LD in different groups

HE staining of the LD of lamb in different groups, the cross-section staining results showed that the muscle fibers were polygonal, all arranged tightly, the muscle bundle gap was large, and the intramuscular membrane was obvious (Figs. 1A–1D). The longitudinal section shows that the muscle fibers are long fusiform, and the nuclei are blue elliptical or rod-shaped (Figs. 2E–2H). From Figs. 1I–1J and Table S3, the difference between the three groups of muscle fiber's area and diameter 0%, 18% and 100% was significant ($P < 0.05$), while the difference between 6% and 18 was not significant ($P > 0.05$), but 18% was less than 6%, indicating that with the increased in the amount of *B. papyrifera* fermented feed the muscle fiber's area and diameter was decreased.

## Data analysis from RNA-seq

In view of the influence of different proportions of *B. papyrifera* fermented feed on the meat quality of fattening lamb, transcription level changes in the LD were analyzed using

**Table 4** Analysis of total fatty acid content in LD muscle (g/100 g).

| Fatty acid | 0% | 6% | 18% | 100% | P-value |
| --- | --- | --- | --- | --- | --- |
| SFA | 49.21 ±0.46b | 50.17 ±0.63b | 49.52 ±0.77b | 52.61 ±0.40a | 0.000 |
| UFA | 50.79 ±0.46a | 49.20 ±0.26b | 50.44 ±0.76a | 46.79 ±0.25c | 0.001 |
| MUFA | 44.17 ±0.60a | 41.12 ±0.52b | 42.97 ±0.23b | 37.09 ±1.31c | 0.000 |
| PUFA | 6.63 ±0.35b | 7.46 ±0.47ab | 8.09 ±0.54ab | 9.70 ±1.32a | 0.015 |
| M/S | 0.90 ±0.04 | 0.82 ±0.03 | 0.87 ±0.02 | 0.74 ±0.05 | 0.194 |
| P/S | 0.13 ±0.01b | 0.16 ±0.02b | 0.15 ±0.03b | 0.18 ±0.05a | 0.009 |
| n3 | 0.20 ±0.02b | 0.32 ±0.02b | 0.23 ±0.02b | 0.79 ±0.19a | 0.003 |
| n6 | 6.08 ±0.35 | 6.64 ±0.65 | 7.58 ±0.46 | 7.99 ±1.44 | 0.511 |
| n6/n3 | 31.65 ±2.75a | 21.70 ±2.85b | 31.78 ±2.83a | 12.95 ±1.23b | 0.000 |

**Notes.**

SFA, saturated fatty acids; UFA, unsaturated fatty acid; MUFA, monounsaturated fatty acids; PUFA, polyunsaturated fatty acids; M/S, monounsaturated fatty acids/saturated fatty acids; P/S, polyunsaturated fatty acids/saturated fatty acids; n3, n3 polyunsaturated fatty acids; N6, n6 Polyunsaturated fatty acids; N6/n3, n6 polyunsaturated fatty acid/n3 polyunsaturated fatty acids.

Different lowercase letters in the same industry indicate significant differences ($P < 0.05$).

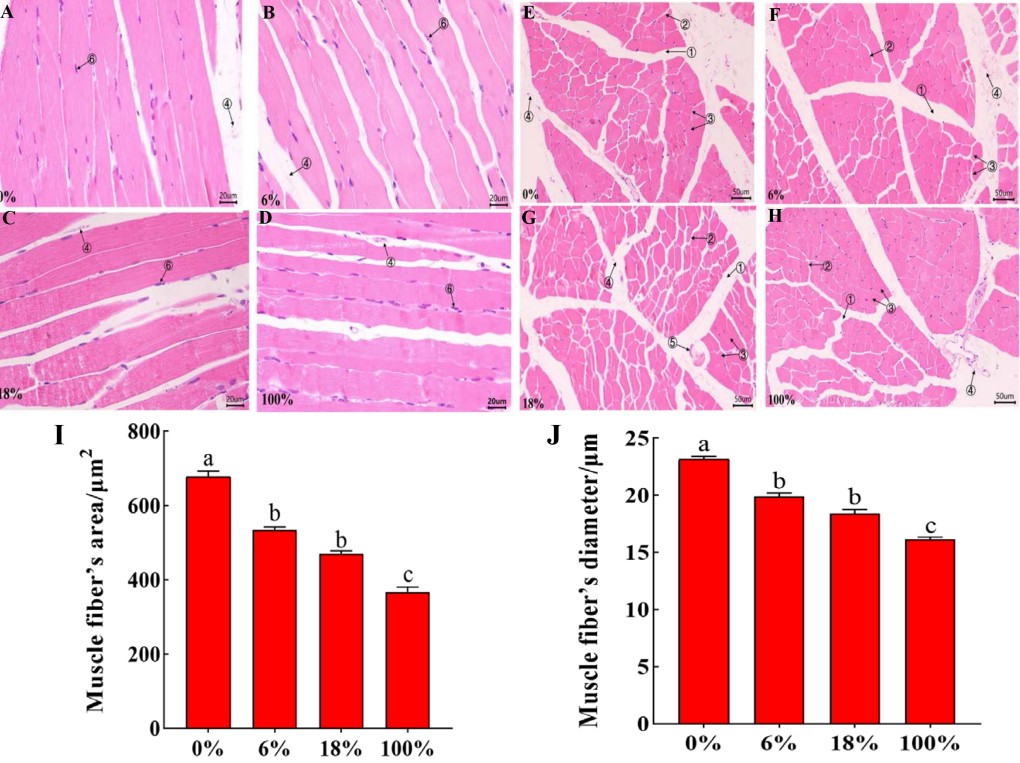

**Figure 2** **Morphological characteristics of H&E between the LD of the sheep with different B. papyrifera fermented feeds.** (A–D) The morphology of the longitudinal view of LD under microscope at 40 ×times magnification; (E–H) the morphology of the transverse view of LD under microscope at 20 ×magnification. (1) Epimysium, (2) endomysium, (3) muscle fiber, (4) blood vessel, (5) nerve, (5) the cell nucleus. (I and J) Muscle fiber's diameter and area in different groups.

**Table 5  The alignment statistics result with the reference gene for all samples.**

| Samples | Clean reads | Clean bases | GC Content | % ≥Q30 | Mapped Reads | Uniq Mapped Reads |
|---------|-------------|-------------|------------|--------|--------------|-------------------|
| 0% | 24,777,328 | 7,408,041,508 | 52.00% | 93.85% | 48,003,609 (96.87%) | 41,605,368 (83.96%) |
| 0% | 25,135,312 | 7,514,998,722 | 51.72% | 93.65% | 48,766,426 (97.01%) | 41,325,592 (82.21%) |
| 0% | 20,761,096 | 6,203,482,124 | 52.00% | 93.57% | 40,190,518 (96.79%) | 35,283,301 (84.97%) |
| 6% | 27,093,623 | 8,116,866,124 | 51.57% | 94.78% | 52,862,303 (97.55%) | 45,396,003 (83.78%) |
| 6% | 26,301,881 | 7,853,198,780 | 51.97% | 95.52% | 51,254,489 (97.44%) | 43,515,259 (82.72%) |
| 6% | 26,594,668 | 7,930,975,080 | 51.25% | 95.64% | 51,935,621 (97.64%) | 43,423,485 (81.64%) |
| 18% | 20,815,698 | 6,220,607,222 | 51.89% | 94.32% | 40,451,204 (97.17%) | 33,974,223 (81.61%) |
| 18% | 21,106,226 | 6,303,522,828 | 51.31% | 93.97% | 40,921,787 (96.94%) | 33,937,852 (80.40%) |
| 100% | 20,274,560 | 6,070,610,678 | 51.57% | 95.16% | 39,563,205 (97.57%) | 33,684,553 (83.07%) |
| 100% | 25,188,343 | 7,525,791,384 | 52.38% | 95.41% | 49,093,882 (97.45%) | 43,162,872 (85.68%) |
| 100% | 24,637,084 | 7,352,446,008 | 52.99% | 94.59% | 47,927,686 (97.27%) | 42,256,798 (85.76%) |

**Notes.**

Clean reads,  the number of clean reads, the single-ended meter;  Clean bases,  the number of clean data; GC content: the percentage of GC-content in clean data; ≥Q30, Q-score of clean data; ≥30, Mapped reads: the number of reads mapped to the reference genome and its percentage in clean reads;  Uniq mapped reads,  the number of reads mapped to the only location of the reference genome and its percentage in clean reads.

the Illumina HiSeq 2000 platform, and prepared to construct cDNA libraries for RNA-seq. As shown in Table 5, After filtering, the clean data of each sample reached 6.07 GB, the percentage of Q30 bases was greater than 93.57%. The percentage of clean reads that mapped into the sheep reference genome ranged from 96.79 to 97.57%. About 80% of clean reads were uniquely mapped and used for subsequent analysis.

## Gene expression analysis

Comparing the three groups to yield a total of 443 DEGs, among them, 48 (19 up-regulated and 29 down-regulated), 104 (38 up-regulated and 66 down-regulated) and 198 (84 up-regulated and 141 down-regulated) DEGs belonged to 6%,18% and 100%, respectively. The 6 DEGs (2 up-regulated and 4 down-regulated) were commonly regulated by effecting of fermented feed from *B. papyrifera* in the three groups (Fig. 3A). A total of 89, 157 and 269 DEGs were identified in the groups of 0% vs 6%, 0% vs 18%, and 0% vs 100%, respectively (FC > 1.5, FDR < 0.05). After adding 6%, the number of down-regulated DEGs (52) were more than the number of up-regulated DEGs (37). When fed all *B. papyrifera* fermented feed, the number of up-regulated DEGs increased from 37 (6%) to 98 (100%), and the down-regulated DEGs increased from 52 (6%) to 171 (100%) (Fig. 3B). Hierarchical clustering represents the difference and similarity of 443 DEGs (Fig. 3C and Table S4). The results showed that there was a significant difference in gene expression profiles differences among the four groups. With the increase of the added amount, the genes with high expression at 0% were gradually down-regulated, while those with low FPKM at 0% were gradually up-regulated. It indicated that the addition of *B. papyrifera* fermented feed a set of gene expression or induces another set of gene expression, although these changes were to regulate the meat quality performance.

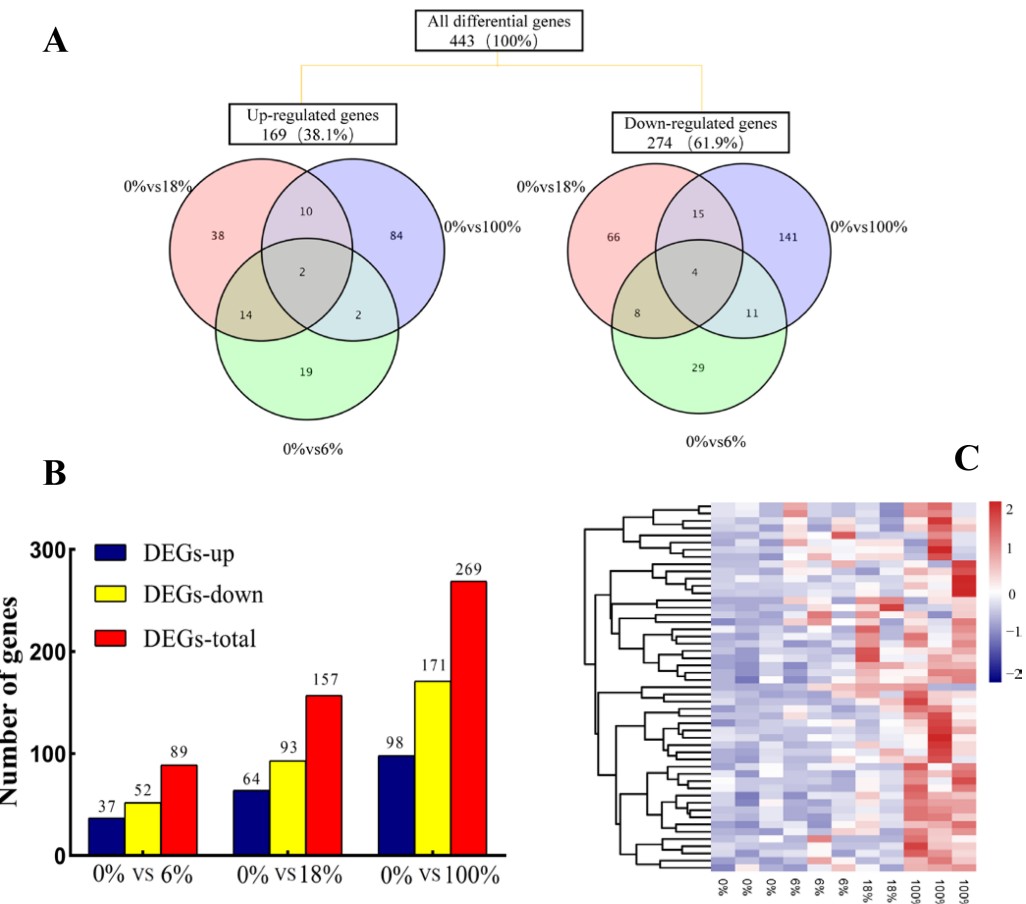

**Figure 3  The expression profile of differentially expressed genes (DEGs) of sheep's LD after adding different B. papyrifera fermented feed.** (A) Venn diagrams representing the numbers of DEGs and the overlaps of sets obtained across three comparisons. (B) Column diagram representing the numbers of DEGs in three groups. (C) The heat map representing the expression profile of 50 of the 443 DEGs.

## Expression pattern and functional analysis of the DEGs in LD of sheep

According to the expression profiles, 443 DEGs were classified into 8 clusters by co-expression clustering (Fig. 4 and Table S5). A total of 16 and 67 genes were classified as clusters 4 and 6, which were rapidly down-regulated within 0% to 18% and then gradually up-regulated within 100%. Most of them participated in pathways such as "Rap1 signaling pathway", and "protein digestion and absorption", it indicates that those genes were transiently inhibited by *B. fermenta*. While the genes in clusters 1 and 5 were up-regulated from 0% to 18%, and gradually down-regulated within 100%. This result indicates that those genes were transiently promoted by *B. fermenta*. Most of them participated in pathways such as "glucagon signaling pathway", "PPAR signaling pathway" and "fat digestion and absorption". The genes belonging to clusters 2, 7 and 8 were continuously down-regulated. Transcription is inhibited by B. fermented feed, most of which are rich in "MAPK signaling pathway" and "cAMP signaling pathway" pathways. Cluster 3 contains

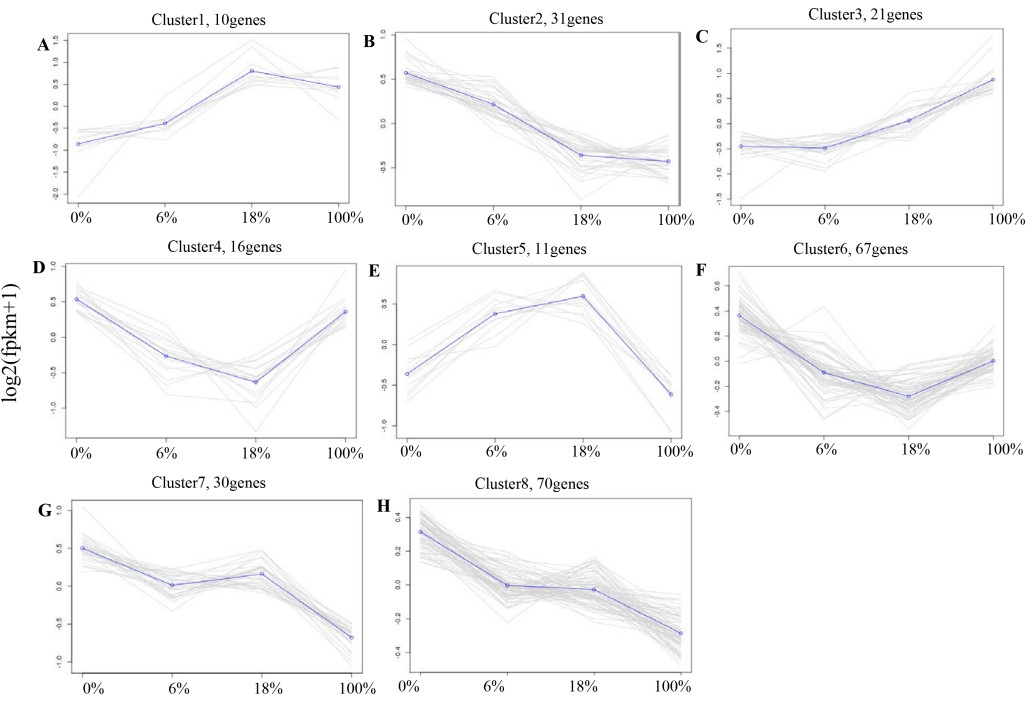

**Figure 4** **A-H) Co-expression clustering, showing the expression profile of 443 DEGs.** The *X* axis represents the amount of different Broussonetia papyrifera fermented feed (0, 6, 18 and 100%). The *Y* axis represents the value of the relative expression level (log2 (FPKM + 1)).

fewer genes (31), and the expression levels of these genes are significantly up-regulated with the addition of fermented feed from *B. papyrifera,* most of which are rich in the ''AMPK signaling pathway'' and ''Circadian rhythm'' pathways.

## GO classification analyses

Annotated genes were divided into three major functional categories: biological processes (BP), cellular components (CC) and molecular functions(MF). It is obvious that there were more functional terms for BP and relatively few transcripts for CC and MF. Compared with down-regulated transcripts, detoxification (GO: 0098754), synapse (0045202), synapse part (0044456) antioxidant activity (GO:0016209) terms were peculiar in up-regulated DEGs; nucleoid (0009295) was unique in upregulation GO terms (Figs. 5A, 5B and Table S6).

## KEGG annotation analyses

KEGG pathway analysis was conducted to investigate whether the genes in LD of the sheep participate in some special pathways with the addition of different *B. fermenta.* fermented feed. Top20 of KEGG enrichment showed that up-regulated DEGs were highlighted in ''fatty acid degradation (ko00071)'', ''circadian rhythm(ko04710)'', ''tryptophan metabolism (ko00380)'', ''arginine and proline metabolism (ko00330)'', ''regulation

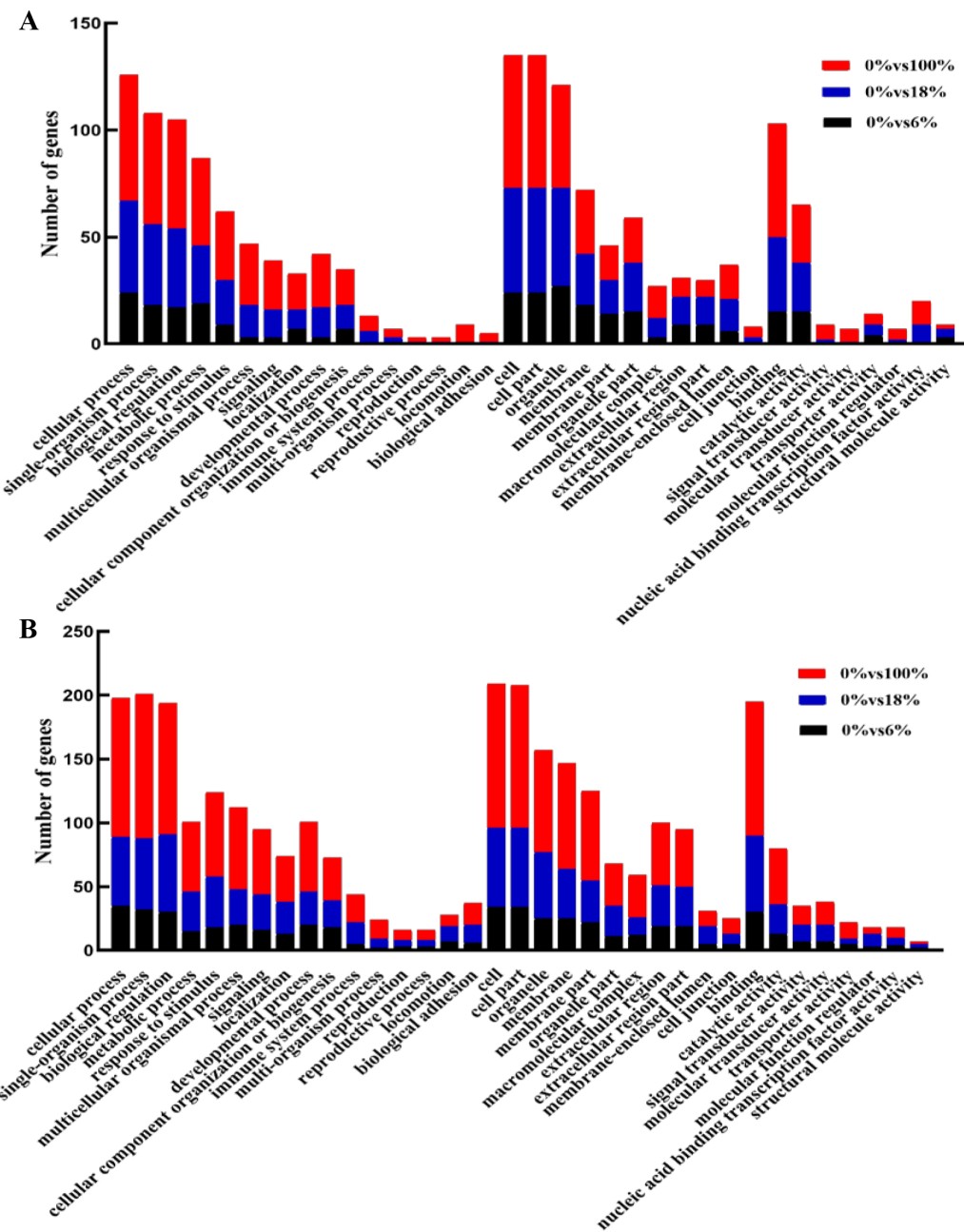

**Figure 5** **GO functional annotation of the differentially expressed genes (DEGs) of sheep's LD in different groups.** (A) Up-regulated DEGs, (B) down-regulated DEGs.

of lipolysis in adipocytes (ko04923)'', ''glycerolipid metabolism (ko00561)'' and ''cGMP-PKG signaling pathway (ko04022)''. Moreover, ''pertussis (ko05133)'', ''proteoglycans in cancer (ko05205)'', ''complement and coagulation cascades (ko04610)'', ''ECM-receptor interaction (ko04512)'', ''pathways in cancer (ko05200)''and ''Rap1 signaling pathway (ko04015)'' annotated the most down-regulated DEGs (Figs. 6A, 6B, and Table S7).

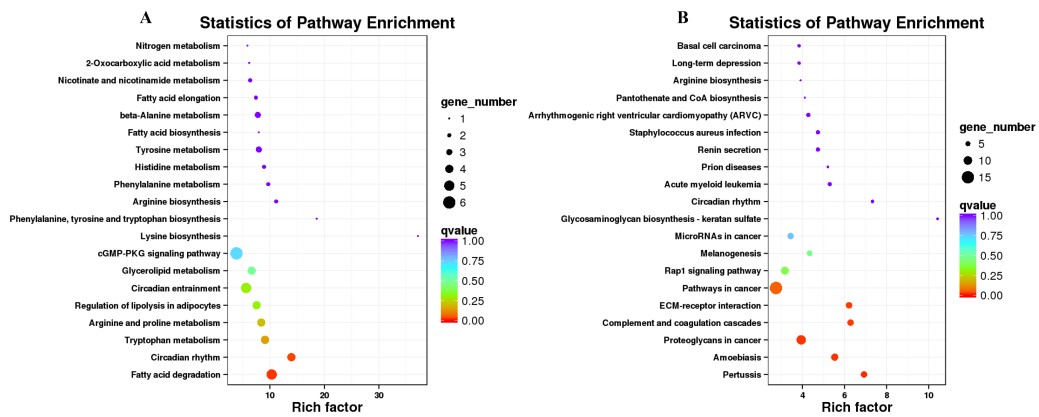

**Figure 6** KEGG pathway enrichment of the differentially expressed genes (DEGs) of sheep's dorsal longest muscle in different groups. (A) Up-regulated DEGs, (B) down-regulated DEGs.

## RNA-Seq expression validation by qRT-PCR

As shown in Fig. 7 and Table S8, we selected 9 DEGs that were closely associated with the meat quality. The result showed that qRT-PCR expression patterns were consistent with the changing trends from RNA-seq data.

## PPI network

The PPI network of DEGs related to meat quality in the LD muscle of lamb were presented in Fig. 8. There were 43 edges and 17 nodes in the network. Based on the PPI natwork, *ACSL1* (degree = 9), *ACADVL* (degree = 9), *ECI1* (degree = 9), *ECHS1* (degree = 9) were the top four hub genes.

## DISCUSSION

### Effect of *B. papyrifera* fermented feed on meat quality traits

The internal quality of meat is dependent on certain characteristics of muscle fibers. Besides, there are many factors that affect the differences of muscle fibers, such as growth and development stage, gender, environment, and nutrition. The size and number of muscle fibers are important factors that affect meat quality. Therefore, the characteristics of muscle fibers are of great significance to the meat quality characteristics and growth of animals (*Shen et al., 2014*; *Kim et al., 2018*). Muscle bundles were composed of muscle fibers, and the size of muscle fiber's diameter is also related to the shearing force. The smaller the muscle fiber's diameter, the tenderer the meat quality. At the same time, the cross-sectional area of the muscle fiber was also one of the important indicators for judging the quality of the meat (*Bidanel et al., 1991*).In this study, the diameter and area of muscle fibers decreased with the increase of the ratio of *B. papyrifera* fermented feed (Refer to Fig. 2), indicating that the addition of *B. papyrifera* fermented feed was directly proportional to muscle tenderness. Therefore, this feed can be added as a protein feed in ruminant diets for improving muscle tenderness.

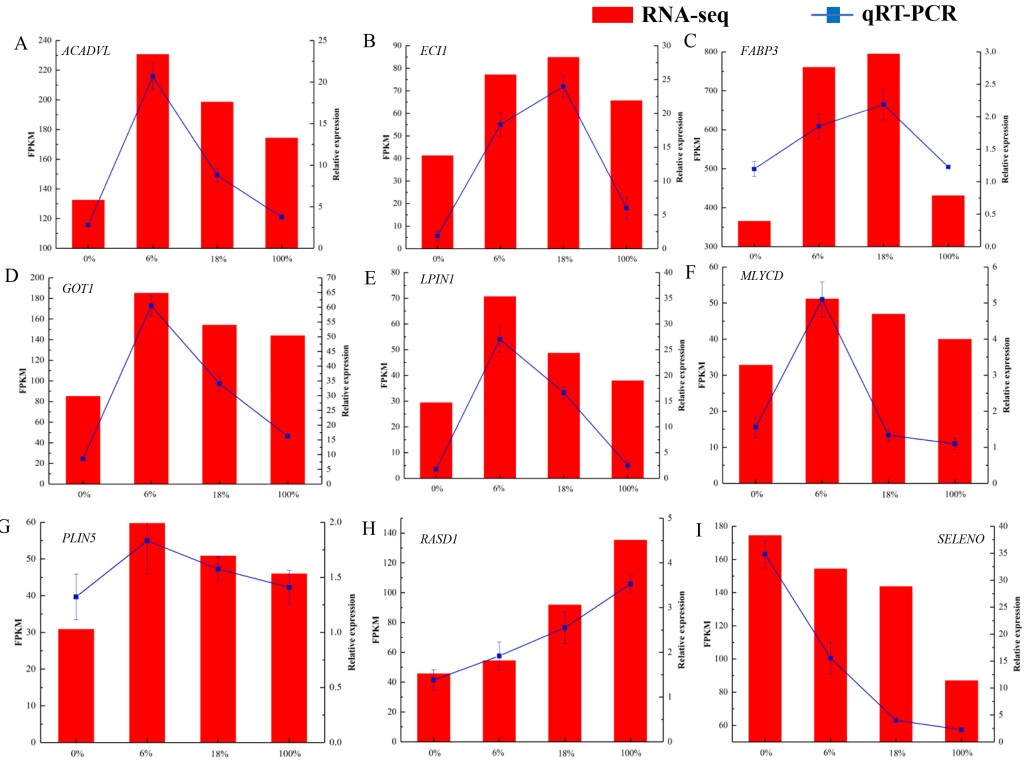

**Figure 7** **(A-I) qRT-PCR analysis of selected DEG genes in the longissimus dorsi of sheep fed with different Broussonetia fermented feed.** Histogram represent the relative expression level defense by qRT-PCR (right *y*-axis). Broken line indicate the change in transcript level according to the FPKM value of RNA-seq (left *y*-axis).

Lauric acid (12:0), myristic acid (14:0) and palmitic acid (16:0) in long-chain SFA are closely related to cholesterol and can harden human blood vessels, which was the reason to cause cardiovascular disease and arteriosclerosis, there are some epidemiology showing that stearic acid (C18:0) is related to coronary heart disease (*Hu et al., 1999*; *Hunter, Zhang & Kris-Etherton, 2010*). In this study, the content of lauric acid (12:0) and myristic acid (14:0) in the group100% was higher than other groups (Refer to Table 2), indicating that excessive addition of *B. papyrifera* fermented feed in the ruminant feed would cause long chains SFA increase. The intake of unsaturated fats can effectively reduce the level of cholesterol in the body and prevent atherosclerosis (*Cameron et al., 2000*; *Wood et al., 2008*). PUFA have been proven to prevent and treat cardiovascular diseases, and also have important physiological effects such as delaying aging and anti-cancer. MUFA plays an important role in lowering cholesterol (*Harris, Poston & Haddock, 2007*). The n3 series of PUFA play a variety of functions in the human body. They can reduce triglycerides in blood lipids, improve blood vessels and eliminate inflammation, and reduce the morbidity and mortality of cardiovascular disease patients (*Philip & Calder, 2013*). In the present study, PUFA were higher in other three groups than that in control group, and it might due to the antimicrobial activities in the diet supplemented with *B. papyrifera* fermented feed

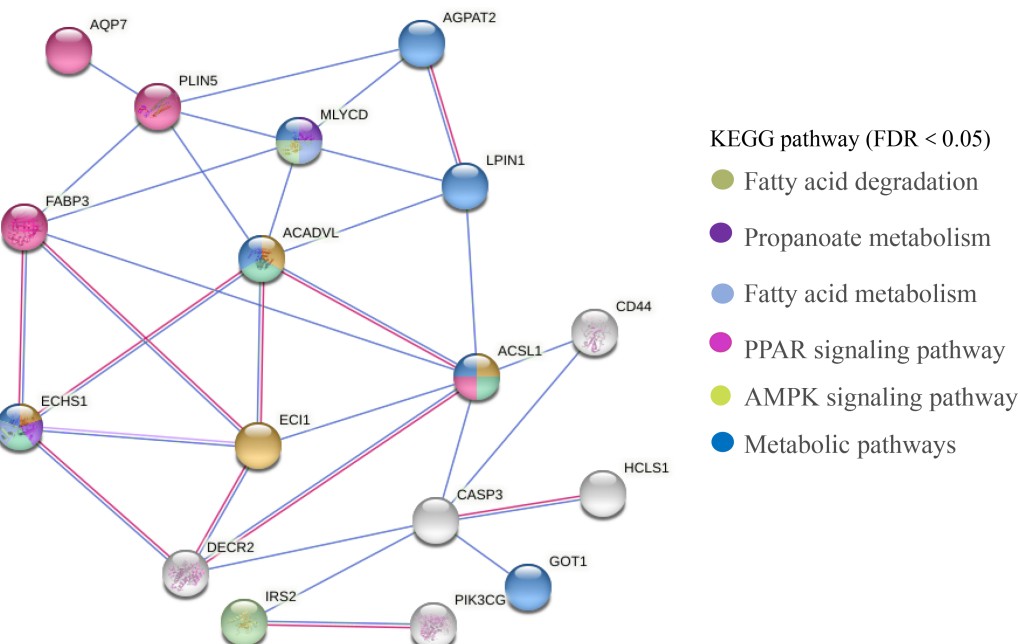

**Figure 8** **PPI network analysis of identified DEGs associated with fatty acids.** Line color indicates the type of interaction evidence.

(Refer to Table 4), as it is reported that prenylated flavonoids isolated from *B. papyrifera* have antimicrobial activity (*Sohn et al., 2004; Sohn, Kwon & Son, 2010*), which affects the biohydrogenation of unsaturated fatty acids by ruminant rumen microorganisms, and also dietary antioxidant activity might increase PUFA content in meat (*Sanchez-Muniz et al., 2012; Bellés et al., 2019* ). Combining the content of beneficial fatty acids such as DHA(C22:6n3), linoleic acid(C18:2n6c) and $\alpha$-linolenic acid(C18:3n3) and harmful fatty acids such as lauric acid(C12:0), myristic acid(C14:0) and stearic acid(C18:0) in the 4 groups, it showed that the 18% group has a higher content of beneficial fatty acids and a lower count of harmful fatty acid, which would be more beneficial for the consumer's health.

## Effects of *B. papyrifera* fermented feed on meat quality-related DEGs and fatty acid metabolic pathways

In this study, we screened 443 genes whose mRNA abundance changes not less than 1.5 times based on FDR (Refer to Fig. 3). Intramuscular fat is one of the indicators reflecting meat quality and flavor, and is closely related to carcass and meat quality. *B. papyrifera* fermented feed mainly improves the intramuscular fat content of fattening lamb through nutritional regulation, thereby affecting meat quality performance. By analyzing the KEGG pathway of differentially expressed genes, it can be seen that 4 DEGs (*ACSL1, AQP7, FABP3* and *PLIN5*) are enriched in the PPAR signaling pathway (Refer to Fig. 8). FABP3 mainly regulates the body's fat and glucose balance in the PPAR signaling pathway, and plays an important role in the transportation and metabolism of fatty acids in cells. It can

transport long-chain fatty acids from the cell membrane to fatty acid oxidation, triglyceride and phospholipid synthesis (*Veerkamp & Maatman, 1995*). *FABP3* gene mutations and mRNA expression levels have a significant impact on intramuscular fat, which in turn affects muscle tenderness (*Wang et al., 2015*). *ACSL1* is the most important synthetase for acyl-CoA to synthesize triglycerides, which helps fatty acid transport and triglyceride deposition (*Richards et al., 2006*). *AQP7*, a water/glycerol transporting protein, regulates adipocyte glycerol efflux and influences lipid and glucose homeostasis (*Oikonomou et al., 2020*). *PLIN5* is expressed on both lipid droplets and mitochondria, and may participate in the interaction between lipid droplets and mitochondria (*Wang et al., 2013*). Muscle-specific overexpression of *PLIN5* increases the storage of lipid droplets in muscle cells and also increases the rate of oxidative gene expression and metabolism (*Harris-Ann et al., 2015*). The formation of larger lipid droplets with a higher degree of esterification, increased methylene content and more saturated lipids (*Nils et al., 2015*). The above results indicate that these four genes mainly promote the deposition of intramuscular fat by participating in the synthesis of triglycerides and polyunsaturated fatty acids, thereby changing muscle tenderness. In this study, the expression of *ACSL1, AQP7, FABP3* and *PLIN5* genes increased with the addition of *B. papyrifera* fermented feed, and the expression of all genes decreased in 100% group (Refer to Fig. 7), indicating that 18% amount of added *B. papyrifera* fermented feed can improve meat quality.

By analyzing the KEGG pathway of differentially expressed genes, it could be seen that there were four genes (*ECHS1, ACSL1, ACADVL* and *ECI1*) that were significantly enriched in the fatty acid metabolism signaling pathway (Refer to Fig. 8). *ECHS1* in cells could activate mammalian target proteins, and animal target proteins may enhance LPL expression through the PPAR pathway. LPL was involved in the metabolism and transport of lipids, and mainly hydrolyzes the chylomicrons and very low-density lipoproteins present on the surface of capillary endothelial cells to produce fatty acids (*Li et al., 2014*). *ACADVL* mainly catalyzes the oxidation of fatty acids to form C 2-C 3 double bonds, and usually catalyzes C 16-acetyl COA or even longer chain fatty acids (*Aoyama et al., 1995*). The long-chain fatty acyl COA is transported to the mitochondria, firstly binds to *VLCAD*, and interacts with the mitochondrial trifunctional protein. After 2-3 cycles of oxidation, the medium-chain fatty acyl CoA is formed (*Houten & Wanders, 2010*). The protein encoded by *ECI1* was an important mitochondrial enzyme involved in the β oxidation of unsaturated fatty acids, which was mainly involved in the metabolic process of fatty acids (*Williams & Tjian, 1991*). In this study, the expression level of *ECI1* increased with the addition of *B. papyrifera* fermented feed, reaching the highest in the 18% group, but decreased in the 100% group. The expression level of *ACAVDL* and *ECHS1* rose to the highest after 6% *B. papyrifera* fermented feed, and then decreased with the increase of the added amount (Refer to Fig. 7). It showed that adding the appropriate amount of *B. papyrifera* fermented feed would increase the synthesis of long-chain fatty acids, while excessive addition of *B. papyrifera* fermented feed would increase fatty acid metabolism and reduce fatty acid content in the body, thereby affecting the fat deposition on meat quality in the LD of lamb.

## CONCLUSION

Although there are many researches on *B. papyrifera* in animal breeding, they have focused on its antioxidant activity, rumen hydrogenation and its effect on fatty acids. There is no report on the specific influence mechanism of *B. papyrifera* fermented feed on the meat quality traits of animal. To our best knowledge, this is the first report about the effect of adding different *B. papyrifera* fermented feed on the meat quality traits and changes of the longissimus dorsal transcriptome of fattening lamb. First, the fatty acid content and muscle fiber morphology of the longissimus dorsal were studied. It was found that after adding *B. papyrifera* fermented feed, the muscle fiber's diameter of the muscle became smaller, the tenderness became more tender, and the content of polyunsaturated fatty acid was the highest in the group18%. Finally, a total of 443 DEGs were identified in the four groups. We screened some DEGs that related to synthesis of fatty acids, most of which were up-regulated in the 6% and 18% groups, and down-regulated in the 100% group. Our results show that proper addition of Pichia fermented feed can help the synthesis of long-chain unsaturated fatty acids and the deposition of intramuscular fat. Therefore, it is recommended to add 18% groups, *B. papyrifera* fermented feed to lamb feed to improve meat quality traits.

## ACKNOWLEDGEMENTS

Thanks to Zhongtian Sheep Industry Co., Ltd. for providing us with the lamb breeding base, and BMK Cloud (http://www.biocloud.net) for supporting in data analysis of this study.

### Funding

This work was supported by the National Key R&D Program of China (2018YFD0502100). The funders had no role in study design, data collection and analysis, decision to publish, or preparation of the manuscript.

### Grant Disclosures

The following grant information was disclosed by the authors:
National Key R&D Program of China:  2018YFD0502100.

### Competing Interests

The authors declare there are no competing interests.

### Author Contributions

- Xuejiao An performed the experiments, analyzed the data, prepared figures and/or tables, authored or reviewed drafts of the paper, and approved the final draft.
- Shengwei Zhang performed the experiments, authored or reviewed drafts of the paper, and approved the final draft.

- Taotao Li and Xia Wang analyzed the data, prepared figures and/or tables, and approved the final draft.
- Nana Chen performed the experiments, analyzed the data, prepared figures and/or tables, and approved the final draft.
- Baojun Zhang conceived and designed the experiments, performed the experiments, authored or reviewed drafts of the paper, and approved the final draft.
- Youji Ma conceived and designed the experiments, authored or reviewed drafts of the paper, and approved the final draft.

### Animal Ethics

The following information was supplied relating to ethical approvals (i.e., approving body and any reference numbers):

All experiments involving animals were reviewed and approved by the Animal Committee of Gansu Agricultural University (GSAU-2019-76).

### Data Availability

Data are available in the Supplemental Files. The sheep mRNA sequences are available at the SRA: PRJNA660919, Biosample: SAMN15963870–15963881.

### Supplemental Information

Supplemental information for this article can be found online at http://dx.doi.org/10.7717/peerj.11295#supplemental-information.

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
