# Peer review of "Transcriptomics analysis reveals the effect of Broussonetia papyrifera L. fermented feed on meat quality traits in fattening lamb"

_PeerJ, doi:10.7717/peerj.11295_

## Round 0.1 · original submission · Major Revisions

Reviewers have considered your work to have merit and have recommended areas of improvement. Kindly attend to all queries raised, and I look forward to your revised manuscript.

Reviewer 1 ·

Basic reporting

no comment

Experimental design

no comment

Validity of the findings

no comment

Additional comments

In this paper, the author investigated the effect of adding B. papyrifera fermented feed to the sheep meat quality. The fatty acid content and muscle fiber morphology of the longissimus dorsal transcriptome have been also characterized. The author concluded that the muscle fiber became smaller, tender, and more polyunsaturated fatty acids when added 18% of B. papyrifera in the feed.

Questions:
• [line 110] About the description of GC method for fatty acid analysis, it would be helpful to mention which GC detector was used in the experiment.
• In the figure 1, the annotations inside Fig1 are too small and vague. Please re-make the Fig1.
• In the figure 2, note that (B) is the heat map and (C) is the column diagram, please make some changes based on your figure. Besides, please also put (B) after (A).
• In the figure 4, the words below the x-axis are too small for readers.
• The Figure 1, 3, 4 and 5 will become very vague after zoom-in. It would be very helpful to improve the resolution of these figures. It will help readers to look into more detailed information.

Reviewer 2 ·

Basic reporting

The manuscript entitled “Transcriptomics analysis reveals the effect of Broussonetia papyrifera L. fermented feed on quality traits in fattening lamb” deals with an interesting topic of feeding strategy in the context of feed-food competition. The manuscript need
an English revision from native speaker. The introduction section lacks of previous paper on the same topic applied in ruminants; moreover, some information about the importance of transcriptomic analysis for understanding the molecular mechanisms activated by feeding strategy for meat production have to be included.

Experimental design

The methods have been well-conducted and conceived and give the opportunity of growing the knowledge about the molecular mechanisms involved in feeding strategy on physiology.

Validity of the findings

The results are in line with the objective of the paper; however, one missing point is represented by the best percentage of Broussonetia papyrifera L. fermented feed for fattening lambs considering both positive and negative aspects.

Additional comments

Point to point comments are listed below:

L33: Please change into “INTRODUCTION”.
L48: Please specify the country without “my”.
L60: Please define “HE”
L81: Please correct with “-80°C”
L89: For Zlatkis et al. reference, please add only the year between parentheses.
L97, 98: Please use “FAME”
L100: Add point after filtered.
L164: Please use the same character dimension for “(C4:0, C10:0, C6:0, C18:0, C12:0, C14:0; C20:0), and delete space after “:”. Pay attention to this suggestion along with the manuscript.
L174: Please use acronyms “MUFA” and “PUFA”
L178: The increasing it is not gradually for C17:1 and C20:1, because it increases only with 100% treatment. Please correct this sentence.
L203: Please change “B“ into “B. papyrifera” Pay attention to this suggestion with along the manuscript.
L205: Please substitute “.” with “,”. after “S4”. In my opinion, table S4 has to be included in the manuscript.
L243: Please substitute “B. fermenta” with “B. papyrifera”.
L283: Please substitute “figure 2” with “Figure 7”. Moreover at L284, in the end, one more point has been added. Please delete it.
L304-305: In the discussion section it is strongly recommended do not add more results and P-value. Please correct or delete this sentence.
L313-316: I don’t agree with the increase of PUFA found in the LD muscle and the antimicrobial activity of flavonoids. The authors have to discuss better this point.
L348-350: Which is for the authors the best quantity of B. papyrifera supplementation in order to obtain the best meat quality?


For all tables, a column with P-value has to be added.
Table 2 caption: Please add an "analysis of fatty acids composition in LD muscle".
Figure 1. Poor quality, it is not possible to read the numbers that indicate muscle morphology.
Figure 4: Poor quality of the figure.

---

## Round 0.2 · Minor Revisions

Thank you authors for your efforts to revise your manuscript. Reviewer have considered to accept, however, Editor requires your kind attention to the following observations, before taking a decision:

A) Introduction:
-The rationale/justification for the use of Broussonetia papyrifera L. fermented feed is not clear for this study. That is to say, why Broussonetia papyrifera L. fermented feed? introduction paragraphs 1 and 2, does not answer this question. paragraph 3 neither. Please brainstorm on this, it is important.
- Create a new paragraph 3, you have to discuss establish why meat quality traits are important. What are meat quality trait? Why is it relevant in fattening lamb? Then, follow it by what is Transcriptomics analysis. What does it entail, what processes are involved, and what is its scope.
- then, the current paragraph 3, would become paragraph 4. Please note line 55, error 'measuring' not 'measuringe'. Make sure your objective statement is clear. Make sure your rationale, specific to this study is clear.

B) Materials and methods:
-Please authors, considering the somewhat rigorous steps in this work, Editor encourages that materials and methods must start subsection titled 'overview of experimental program'. make sure you provide a schematic diagram, that shows the step wise sequence of the entire study. The essence of this section is to provide the reader with a snapshot of what you have done. This section should have 3-4 sentences.
Design of study>Assembly of experimental animals> Transition/Pre-test/Formal Periods >Lamb preparation (schematize how lamb were allocated for analysis) > Analytical methods (Schematize the various analyses)
- After this section, since chemicals were used, next subsection should be 'chemicals and reagents', please, kindly outline the chemicals used, and their sources.
- After this section, next subsection is 'Ethics approval'
The sentence of lines 61-62, should be put here, so that it stands alone. Any other information, that pertains to ethics should be here.
-Next, will then be 'Experimental animals and design', which should be changed to be: 'Preparation of experimental animals'. It is very interesting that method of Zlatkis et al(1953) was used. Please, kindly add a sentence why this method stands unique.
- Please, change 'Data statistics and analysis' to 'Statistical analysis'. Lines 134-135, please amend to the following:
-- one-way analysis of variance (ANOVA) was used to analyse the emergent data.
--least significant difference (LSD) was used to resolve the mean differences
(which least sgnificant difference method was used? Is it Fischer's? or? please, clarify)
--level of probability was set as follows: p<0.05 considered statistically significant, and p<0.01 considered statistically extremely significant
--SPSS 22.0 software (SPSS Inc., Chicago, IL, 134USA) was used to do the data analysis
{no need for 'The data analyses method adopted were'}
kindly oblige to follow it this way
??why did you separate the p<0.05 and p<0.01.
It is still ok, however, make sure you kindly provide the R-sq values in Tables 2,3, and 4 (in the 7th column)
Because you have separated it, make sure you provide indicate the exact p values in all the places in the results section where p<0.05 and p<0.01 (so, it is not just saying p<0.05, replace it with (p=0.??, R-sq=??)

Results
-Line 238 (shift 'PPI network' down)
This section is ok as it is

Discussion
- Please, go through the entire two subsections of the discussion. Kindly insert, (Refer to Table ??? or Refer to Figure ??) at all the places where the results of Table or Figure has been referred to. Please, Editor will be looking out for these.

- Line 245 'The meat quality of muscles ...' does not seem right, do you mean to say 'The internal quality of meat is dependent on certain characteristics of muscle fibers'??
Because you have raised an argument, the next sentence should read.. 'Besides, there are many factors that affect the differences of muscle fibers, such as growth and development stage, gender, environment, and nutrition.
-Line 274 ... antimicrobial activity (Sohn et al. 2004; Sohn et al. 2010), which affects.....

- Line 325 ... It showed that adding a proper amount of... ' please change it to ' It showed that adding the appropriate amount(s) of ....

- Line 330, you have made this claim.... This is the first report about the effect
That is the reason why , at the introduction, you have to make a very strong justification, why this is the first study of this kind.
Please, editor encourages authors to fortify the introduction of this work with substantial details here requested . Editor will look very closely to examine this in the revised manuscript.
In this conclusion, why is it the first study? and how is it the first study?

Thank you very much for this very brilliant study. Editor encourages authors to kindly attend carefully to the revision. Look forward to recieving your revised manuscript.

Reviewer 1 ·

Basic reporting

The author has made changes on the grammar and language. Now the manuscript is clear and unambiguous.

Experimental design

This research meets the aims and scope of PeerJ journal.

Validity of the findings

All the data and results in this manuscript have been provided and robust, statistically sound and controlled.

Additional comments

The author has answered my questions and comments. I would like to support this manuscript to publish at PeerJ. Thanks.

---

## Round 0.3 · Minor Revisions

Thank you for revising the manuscript, kindly attend to the following:

- Please, where is the schematic diagram, to demonstrate the overview of the experimental program? Kindly develop a figure, that demonstrates this. Please, kindly construct this subsection, compulsorily with few sentences to make meaning, to readers. Figure 1 shows the schematic diagram of this, from this, to this. Then, tell us why each key stage is relevant to achieving the objective of this current study, very briefly. This is why this is called the overview of the experimental program.
- In the entire 'discussion', where are '(Refer to Table ? or Fig ?') to help readers connect with the aspect of the result that is being discussed? Please, carefully go through the entire discussion section, and insert (Refer to Table ?) or (Refer to Fig. ?) in all the places where information of specific results of Table or Figure is highlighted/mentioned.
- In your conclusion, you state that this is the first report? Please add, "to our best knowledge, this is the first report...."
Why is this the first report? make the case by differentiating it from what was previously existing in literature, extract this from the discussion, and make this case at the beginning of your conclusion, with two or three sentences.

Please, kindly address these, in your response, as you do so in the manuscript. Thank you for your efforts. Look forward to your revised manuscript.

---

## Round 0.4 · Minor Revisions

Thank you for revising your work. However, the revisions are not completely adequate.

- Please, in the overview of the experimental program, where is the schematic diagram, to demonstrate the flow of the various stages of the entire materials and methods? Kindly develop this figure and relay it by ‘Figure 1 shows the schematic diagram of this, from this, to this’. The information you have provided does not yet tell us why each key stage is relevant to achieving the objective of this current study. Please do this very tersely . This is why this is called the overview of the experimental program.

- The objective statement of this work, at the end of the introduction is not clear cut. Your justification is ok, but the objective statement must stand out.

- The conclusion, it is not yet well tied up, were the objectives achieved? How and why? And connect it better with recommendations directed for future work

Look forward to your revised manuscript

---

## Round 0.5 · accepted · Accept

The authors have done well to revise their work and addressed all concerns raised by reviewers. The authors have benefitted from the peer review process, to arrive at a very improved and revised manuscript, suitable for publication. Thus, this work is now acceptable for publication. Thank you for finding PeerJ as your journal of choice. Looking forward to receiving your future submissions. Congratulations :)